# Spatial Effects of Environmental Regulation and Green Credits on Green Technology Innovation under Low-Carbon Economy Background Conditions

**DOI:** 10.3390/ijerph16173027

**Published:** 2019-08-21

**Authors:** Quan Guo, Min Zhou, Nana Liu, Yaoyu Wang

**Affiliations:** 1School of Management, China University of Mining and Technology, Xuzhou 221116, China; 2School of Business, Global Institute of Software Technology, Suzhou 215000, China; 3Research Center for Smarter Supply Chain, Dongwu Business School, Soochow University, Suzhou 215000, China

**Keywords:** green credit, environmental regulation, green technology innovation, spatial measurement

## Abstract

Based on the data of green credit (GC), environmental regulation (ER) and green technology innovation (GTI) in 30 provinces and cities of China from 2007 to 2016, this study investigated the relationship between green credit and green technology innovation development and analyzed the adjustment effect of ER on GC to promote GTI using Geoda and Matlab2016 software, so as to further guide and encourage GC. The results show that GTI in 30 provinces and municipalities in China has a significant spatial agglomeration effect. Single GC plays a certain role in promoting local technology innovation, but it fails to influences the surrounding areas. Environmental regulation has a certain regulatory effect on the relationship between green credit and green technology innovation in the province but also fails to influences the surrounding areas.

## 1. Introduction

In the process of social and economic development, enterprises play an extremely important role. In today’s era of advocating low-carbon and environmental protection, enterprises have adopted a high-pollution, high-energy development model that clearly cannot meet the needs of low-carbon society development. Moreover, many countries today are faced with the problem of how to solve the economic development under the background of low-carbon environmental protection. Therefore, in the process of China’s economic transformation and development in recent years, enterprises must continue to innovate to meet the needs of low-carbon economic development [1] The characteristics of the low-carbon economy mainly include: First, reducing energy consumption and pollutant emissions, that is, achieving the “three lows” of low energy consumption, low emissions and low pollution in the process of economic development; second, while maintaining sustained growth, reduce emissions of the “three wastes” (waste water, waste gas and waste residues) and improve energy efficiency; third, innovative low-carbon technologies are the most direct means to achieve low-carbon economic development. In recent years, the international community has become increasingly aware of the importance of green innovation as a result of growing global problems such as resource scarcity and environmental degradation caused by rapid economic growth.

Given the social and economic development, as well as policy advocacy, China has experienced a rapid and sudden increase in economic development in the past three decades. Although the economy in China is rapidly developing, numerous problems have also occurred. The extensive growth of China’s economy, affects the ecology and environment and causes natural resource depletion and population explosion; these phenomena potentially hinder “substantial development” [2]. According to the joint report on the Global Environmental Performance Index (GEPI) issued by Yale University Center for Environmental Law and Policy (YCELP), Columbia University Center for International Geoscience Information Network (CIESIN) [3] and the World Economic Forum (WEF), China ranked 94 (40th percentile), 105 (45th percentile), 121 (43th percentile), 116 (17th percentile), 118 (61th percentile) and 109 (72th percentile) in 2006, 2008,2010, 2012, 2014 and 2016. The extensive industrial development model has made China fall into an “environmental pollution–economic development” cycle. In this regard, green technology innovation is the key to get out of this “strange circle” [4].

Innovation is the driving force of economic development, while green embodies the demand for environment protection and resource saving. Green innovation has become an inevitable choice for global societies to realize long-term development and gain competitive advantages over other countries. Since the reform and opening up, with the rapid growth of China’s economy, problems such as resource consumption and environment pollution are becoming more and more serious. The “Outline of the Thirteenth Five-Year Plan for National Economic and Social Development of the People’s Republic of China” (13th Five-Year Plan) clearly pointed out that to solve development problems, concepts of innovation-driven development, coordinative development, green development, opening development and sharing development must be firmly established and stuck to, which emphasizes the importance of innovation-driven and green development in China’s economic and social development. Green innovation, as a combination of innovation-driven and green development concepts, will play a more important role than ever before in the new normal where China’s economic development slows down and resource and environment constraints are serious.

Based on the equator principle, green credit (GC) is an international practice that has no legal binding effect and mainly evaluates and manages the environmental, social and credit risks faced by commercial banks when they embark on project financing decisions. GC starts earlier abroad and has caused the emergence of theoretical researches such as green finance and carbon finance [5]. The development of international green finance provided a reference for China to launch a GC policy, but the current GC policy in China differs from international standards. The current GC in China is essentially a macro-control means that integrates green finance with environmental protection. It is a policy orientation that stimulates and supports environmentally friendly and resource-saving enterprises, projects, etc. and financially restricts and punishes enterprises with high pollution and high energy consumption. At present, while researches on the influence of financial development on technological progress are relatively mature, the relationship between GC and technological progress are rarely reported. Zhang et al. investigated the implementation of GC policies at national and provincial level in China, and the results showed that GC policy had not been fully implemented [6]. Jin and Mengqi found that GC policy has made remarkable achievements in energy saving and gas emission reduction and industrial structure optimization [7]. However, a huge gap exists between reality and expectation in the process of policy implementation, because of the collusion between enterprises and local governments. De la Fuente and Martin [8] emphasized the supervisory role of financial development in the process of entrepreneurial technology innovation. They argued that financial development could reduce supervision costs and enable entrepreneurs to obtain more favorable loan terms, while preferential loan conditions could improve the level of technology innovation activities. GC provides preferential credit support for energy-saving and environmental protection enterprises. According to De la Fuente and Martin’s research ideas, GC can improve the technology innovation ability of enterprises through preferential loan conditions. Aghion et al. analyzed the data of financial development and output level of 71 countries and found that financial development would converge the technology of the whole world to the same level, and technological improvement would promote total factor productivity, thus promoting industrial structure upgrade [9]. Tadesse [10] conducted an empirical study on the panel data of 38 countries and found that financial development improves national technology innovation and thus promotes industrial development by raising productivity.

As to the effect of environmental regulation (ER) on green technological innovation, most scholars have carried out studies in light of the Porter hypothesis without drawing a uniform conclusion. Their conclusions mainly include the following three viewpoints: First, ER promotes the development of green technology innovation (GTI). Jaffe and Trajtenberg [11] held the idea that the cost of ER policy that enterprises stuck to relied on the costs of green production technology and pollution control technology, while the technology itself could also change the nature and speed of technology evolution. Taking the American manufacturing industry as an example, Brunnereier and Cohen [12] found that there was a certain positive correlation between ER strength and industrial environmental patent, and for every $1 million increase in investment in environmental pollution control, the number of patent licenses for industrial environment would increase by 0.04%. Also taking American manufacturing industry as an example, Carrion-Flores [13] drew the conclusion that there was an obvious negative correlation between pollutant emissions and environmental technology patents, and that the implementation of ER policy in the United States is conducive to stimulating GTI in enterprises. Taking European Union as an example, Lanoie et al. [14] believed that ER promotes some environmental innovation, and it can also stimulate innovation to reduce costs under certain conditions. By constructing the panel data model and taking Germany as an example, Horbach [15] analyzed the factors that influenced the innovation of environmental protection. The results show that the changes of environmental control, environmental management tools and general organization can stimulate the development of GTI. According to the research of Lee and Veloso [16], it can be drawn that the higher the level of environmental regulatory standards, the greater the contribution to GTI and the impact on the subsequent technological changes. Lin and Yang [17] analyzed the relationship between environmental regulation and technology innovation in three different regions of China, and the results showed that ER has a positive effect on technology innovation in these regions in the long term. Second, ER hinders the development of GTI. Filbeck and Gorman [18] carried out an empirical analysis of the relationship between the environmental and financial performances of an enterprise. The result shows that there is a negative correlation between ER and the rate of financial return. Wagner [19] constructed negative binomial and binary discrete models and selected the relevant data of German manufacturing enterprises as an example to analyze the relationship between ER, GTI and patent application. It is found that the standard level of ER implementation hinders the patent application activities of enterprises to a certain extent. Chintrakarn [20] constructed a stochastic frontier analysis model to evaluate the impact of ER on technology inefficiency in manufacturing industries in 48 U.S. states. The results show that the more stringent ER in American manufacturing industry, the more significant its positive impact on technology inefficiency. Third, ER has no significant effect on GTI. Jaffe and Palmer [21] analyzed the overall research and development activities of American enterprises. The results show that the relationship between the number of patents and ER policy is not clear. Aipay et al. [22] explored the influence of ER on industrial productivity under different market structures from the perspective of market structure. The results show that there is a negative correlation between ER and the productivity of American food processing industry, but there is no significant relationship between ER and the profit margin of the industry. Guo et al. [23] test whether there is a significant” U-shaped” relationship between ER and GTI, and found that there exists an “inflection point” in the role of environmental regulation in GTI, and China is at the stage of inhibition before the “inflection point”. The strict environmental law caused Chinese listed enterprises to face higher environmental regulation costs, public pressure and environmental litigation and the financing capacity of heavily polluting enterprises has dropped significantly, especially in areas with higher regulatory intensity [24]. The intensity of environmental regulation in different regions has different effects on the development of green technology innovation. Thus, Hu et al. [25] found that in the low discharge regulation scenario, labor-based foreign direct investment (FDI) has a significant negative spillover effect, and capital-based FDI has a significant positive spillover effect. However, in the high-intensity environmental regulation industry, the negative influence of labor-based FDI is completely restrained, and capital-based FDI continues to play a significant positive green technological spillover effects. Feng and Chen [26] examined the role and mechanism of environmental regulation in the impact of green innovation on industrial green development, considering the impact of environmental regulation on industrial green development performance, found that different types of environmental regulation have different regional influences.

According to the literature, although some studies have come to a relatively rich conclusion in the relevant fields and provided a good foundation for promoting GTI in the new period, there are still problems to be further studied. Does GC realize the true sense of GTI? Is there a need to further consider the impact of interaction between GC and ER on GTI? According to existing researches, related researches mainly focus on GC and technology innovation as well as ER and technology innovation, while discussions on the relationships between GC, ER and technology innovation are relatively few. The contributions of this paper are mainly reflected in the following aspects: (1) On the basis of researching GC and GTI, further considering the impact of interaction with environmental regulation on GTI, linking GC, ER and GTI, which has enriched the literature on the factors affecting GTI. (2) From the research method point of view, at present, the research on green technology innovation and its influencing factors mainly adopts the cross-sectional, time series or panel data methods, but these methods regard the research unit as individuals who are independent of each other ignoring spatial correlation. The development of any region cannot rely solely on its own resources, but is increasingly influenced by the surrounding areas. Based on the establishment of spatial econometric model and spatial effects, the spatial measurement method is used to study the impact of green credit and environmental regulation on green technology innovation.

Therefore, based on relevant data of GC, ER and GTI of 30 provinces and cities in China from 2007 to 2016, the paper analyzed the spatial correlation between GC, ER and GTI by using Geoda (Dr. Luc Anselin and his team, Chicago, American) and Matlab2016 (Math Works, Massachusetts, American) to establish a spatial measurement model. In this way, it examined the adjustment effect of ER on GC to promote GTI and revealed the influence of GC on GTI. The study can play a significant role in giving full play to the promotion effect of GC, guiding rational flow of funds and realizing the development of high-quality economy.

The remainder of this paper is organized as follows: Section 2 describes the research methods, which involve sample selection, data collection, model development and the definition of variables. Section 3 present the results and analysis, and Section 4 summarizes our results and concludes with some suggestions.

## 2. Methods

### 2.1. Variable Selection and Data Sources

China has 34 provincial-level administration regions, of which 30 provinces and cities (Tibet, Hong Kong, Macao, and Taiwan are excluded) from 2007 to 2016 are investigated in this study, considering the continuity of index and the availability of data. These 30 provincial-level administration regions can be divided into four groups, namely, eastern, northeastern, central, and western regions. The eastern region comprises 10 administrative areas, including three out of the four municipalities in China, namely, Beijing, Shanghai, and Tianjin. The other seven administrative areas include Fujian, Guangdong, Hainan, Hebei, Jiangsu, Shandong, and Zhejiang Provinces. The northeastern region includes Heilongjiang, Jilin, and Liaoning Provinces, whereas the central region includes Anhui, Henan, Hubei, Hunan, Jiangxi, and Shanxi Provinces. The western region includes four autonomous regions, one municipality (i.e., Chongqing), and six provinces. The autonomous regions are Guangxi, Inner Mongolia, Ningxia, and Xinjiang, and the provinces are Gansu, Guizhou, Qinghai, Shaanxi, Sichuan, and Yunnan [27].

Raw data were obtained from The Statistical Yearbook of China (2008–2017), and some of the missing data were supplemented by the global statistical data/analysis platform The Easy Professional Superior (EPS), The Statistical Communique on National Economic and Social Development of provinces and cities, and The Urban Statistical Yearbook of China. Foreign direct investment was converted to RMB through the exchange rate, and exchange rate data were obtained from The Statistical Yearbook of China. The paper started the research from the data in 2007, mainly because in 30 July 2007, the China Environmental Protection Bureau, the People’s Bank of China and the China Banking Regulatory Commission (CBRC) issued The Opinions on the Implementation of Environmental Protection Policies and Regulations to Prevent Credit Risk (hereafter referred to as The Opinions). The Opinions put forward the terminology green credit for the first time and proposed a series of credit delivery requirements for industries with industries high energy consumption, heavy pollution and overcapacity, representing the official start of GC.

Interpreted variable-GTI: There are many methods to measure GTI (Du et al., [28,29]). The study is based on the availability of data and refers to the views put forward by Vijvers [30], Schmoch [31], etc. The paper takes the corresponding environmental technology field in international patent classification as the selection criterion, measures the level of GTI with the number of invention and utility model patent authorization and takes the natural logarithm. Technology patent is the most important output and index of innovation, so it is reasonable to use the number of green technology patents to measure the development of green innovation. In order to research on green technology patents more conveniently and efficiently, the intellectual property offices in many countries have put forward green technology patent classification index. The World Intellectual Property Organization launched an online tool, The Green List of International Patent Classifications, in 2010 to facilitate the retrieval of patent information related to environmentally friendly technologies. Based on the United Nations (UN) Framework Convention on Climate Change, the retrieval index categorizes green patents into seven categories: transportation, waste management, energy conservation, alternative energy production, administrative regulatory or design aspects, agriculture or forestry, nuclear power generation. According to the above classification criteria, this paper identified and calculated the number of annual green patents of enterprises, and further distinguished the green invention patent and green utility model patent as the core measurement index of green innovation activities of enterprises. Patents reflect innovation ability to a large extent. Patent application trend can display the trend of new technology development, which is an important embodiment of the degree of technology innovation efforts of regional innovation subjects. Compared with patent licensing, it faces no time lag and is less influenced by people. Therefore, it is often used to represent innovative output. Therefore, this paper selected the number of green patent applications to measure the performance of regional green innovation.

Explanatory variable-GC: Because GC has just come into being for a short time. At present, the academic field generally holds that GC proportion and interest expenditure proportion of high-energy-consumption industries can measure GC. The reverse indicator, namely, the interest expenditure proportion of high energy-consuming industries refers to the ratio of the industrial interest expenditure of China’s six large high-energy-consumption industries to the total industrial interest expenditure. According to the classification given by National Development and Reform Commission of China, the six high-energy-consumption industries are chemical raw materials and chemical manufacturing industry, non-metallic mineral products manufacturing industry, ferrous metal smelting and rolling processing industry, non-ferrous metal smelting and rolling processing industry, petroleum processing coking and nuclear fuel processing industry, power and heat production and supply industry. The proportion of interest expenditure of the six high-energy-consumption industries illustrates the development of GC from the reverse side. This paper referred to China Green Financial Report 2014 by Li et al. [32] and measured GC by using 1-the proportion of interest expenditure of six high energy-consuming industries in total industrial interest expenditure. Specific considerations are as follows: First, the current bank credit can be loaned to six high-energy-consumption industries and other more environmentally friendly industries; Second, the current change of interest expenditure in China’s industry is mainly related to the scale of loans, due to the relatively small gap between the loan interest rates of China’s banks; Third, the six high-energy-consumption industries are characterized by overcapacity, high pollution and high consumption, which deviate from national policies in recent years. The proportion of interest expenditure of high-energy-consumption industries reflects the efforts of commercial banks to encourage environmentally friendly enterprises.

ER: ER is an important factor affecting the innovation of green technology. Scholars have two methods to measure ER strength, namely the comprehensive index and the type index. This paper measured ER strength by using the investment in environmental pollution control accounted for GDP and entered the regression equation in the original form. The advantage is that the investment in environmental pollution control is highly correlated with the degree of environmental governance, and is less affected by GTI, which can effectively alleviate the endogenous problems brought by variables. Generally speaking, the higher the ER value is, the stronger the environmental regulation is.

Control variables refer to the existing research achievements [21,33]. The control variables selected in this study include industrial structure (IS), foreign direct investment (FDI) and scientific and technological fiscal expenditure (STF), capital input (lncapital) and labor input (lnlabor). Among them, IS is the ratio of the total value of the tertiary industry to GDP; FDI refers to the ratio of foreign direct investment to GDP; STF is measured by the fiscal expenditure of science and technology to the total fiscal expenditure of various regions; Lncapital is natural logarithm of social fixed asset investment and lnlabor is natural logarithm of employment population.

### 2.2. Spatial Autocorrelation Test

The global spatial autocorrelation test is used to investigate the spatial correlation of the data index of adjacent geographical regions, and the measure index is the global Moran’s I index. The calculation formula is as follows:
(1)$${\mathrm{Moran}}^{\prime }s\;I=\frac{n\sum_{i=1}^{n}\sum_{j=1}^{n}w_{ij}\left(x_{i}-\bar{x}\right)\left(x_{j}-\bar{x}\right)}{\sum_{i=1}^{n}\sum_{j=1}^{n}w_{ij}{\left(x_{i}-\bar{x}\right)}^{2}}$$
where *ҳ_i_* and *x_j_* are the spatial data of region *i* and *j*; and *w_i_*_j_ is the spatial weight matrix. The value of the Moran’s I index generally lies within [−1–1]. If the value is greater than 0, the data characteristics between different regions are spatially similar and positively correlated. If the value equals 0, it indicates that the data characteristics between different regions are independent from each other without spatial correlation. If the value is smaller than 0, it indicates that the data characteristics between different regions are spatially similar and negatively correlated.

The local space autocorrelation test can measure the spatial aggregation property of the locally studied region, and the measurement logic is similar to the global Moran’s I index. In this paper, the local indicators of spatial association (LISA) agglomeration map based on local Moran’s I index is adopted to display the local spatial correlation characteristics. LISA agglomeration map divides the local agglomeration into high-high (HH), low-low (LL), low-high (LH) and high-low (HL), among which LH indicates that cities with low (L) GTI capacity are surrounded by cities with high (H) innovation capacity. The same case applies to the remaining three types.

### 2.3. Model Selection

The paper aims at studying the relationship between GC, ER and GTI. However, a certain degree of spatial interaction generally exists between economic and geographical behaviors in different regions [34]. Therefore, in order to better explore the impact of GC on GTI mechanism and avoid ignoring the deviation caused by space effects, this paper will conduct ordinary linear regression of the constructed model, detect residual autocorrelation and further judge whether the spatial panel metering model should be constructed to analyze the relationship between GC and the level of green technology.

If there is a spatial autocorrelation between regional innovation activities, then spatial dependence needs to be integrated into the measurement model. The spatial lag model (SLM) incorporates the endogenous autocorrelation effect between the variables into the model, while the spatial error model (SEM) incorporates the autocorrelation effect between the error items. LeSage and Ace established the spatial Durbin model (SDM) which can simultaneously integrate the spatial autocorrelations of error items and the interpreted variables and gain more robust estimates theoretically. The three models are set as follows:
(2)$$\begin{array}{l} {\mathrm{lnY}}_{it}=\alpha_{it}+\rho W\times {\mathrm{lnY}}_{it}+\beta_{1}GC_{it}+\beta_{2}lncapital_{it}+\beta_{3}lnlabor_{it}+\beta_{4}IS_{it}+\beta_{5}FDI_{it}\\ +\beta_{6}STF_{it}+\varphi_{i}+\eta_{t}+\varepsilon_{it} \end{array}$$
(3)$$\begin{array}{c} {\mathrm{lnY}}_{it}=\alpha_{it}+\beta_{1}GC_{it}+\beta_{2}lncapital_{it}+\beta_{3}lnlabor_{it}+\beta_{4}IS_{it}+\beta_{5}FDI_{it}+\beta_{6}STF_{it}+\varphi_{i}+\eta_{t}+\varepsilon_{it}\\ \tau_{it}=\gamma W\times \tau_{it}+\varepsilon_{it} \end{array}$$
(4)$$\begin{array}{ll} {\mathrm{lnY}}_{it}=\alpha_{it} & +\rho W\times {\mathrm{lnY}}_{it}+\beta_{1}GC_{it}+\beta_{2}\;lncapital_{it}+\beta_{3}lnlabor_{it}+\beta_{4}IS_{it}+\beta_{5}FDI_{it}\\ & +\beta_{6}\;STF_{it}+\beta_{7}\times GC_{it}+\beta_{8}W\times lncapital_{it}+\beta_{9}W\times lnlabor_{it}\\ & +\beta_{10}W\times IS_{it}+\beta_{11}W\times FDI_{it}+\beta_{12}W\times STF_{it}+\varphi_{i}+\eta_{t}+\varepsilon_{it} \end{array}$$
where α is the constant; ρ and β are the evaluation coefficients; the subscript *i* and t represent the province and time, respectively; *W* is the spatial weight matrix. The strength of the spillover effect, which can be quantified through the distribution of different weights, is influenced by the geographical distance/economic and technological gap between the two regions and other factors. $\varphi_{i}$ and $\eta_{t}$ are the fixed effect and the stochastic effect, respectively; $\varepsilon_{it}$ is the error item.

In order to investigate the regulatory effect of ER on GC to promote the development of GTI, based on Models (1), (2) and (3), this study introduced the concept of $GC_{it}\times ER_{it}$, namely the interacted items of GC and ER, to reflect their common influences on the development of GTI.
(5)$$\begin{array}{c} {\mathrm{lnY}}_{it}=\alpha_{it}+\rho W{\mathrm{lnY}}_{it}+\beta_{1}GC_{it}+\beta_{2}lncapital_{it}+\beta_{3}lnlabor_{it}+\beta_{4}IS_{it}+\beta_{5}FDI_{it}\\ +\beta_{6}STF_{it}+\beta_{7}ER_{it}+\beta_{8}GC_{it}\times ER_{it}+\varphi_{i}+\eta_{t}+\varepsilon_{it} \end{array}$$
(6)$$\begin{array}{c} {\mathrm{lnY}}_{it}=\alpha_{it}+\beta_{1}GC_{it}+\beta_{2}lncapital_{it}+\beta_{3}lnlabor_{it}+\beta_{4}IS_{it}+\beta_{5}FDI_{it}+\beta_{6}STF_{it}\\ +\beta_{7}ER_{it}+\beta_{8}GC_{it}\times ER_{it}+\varphi_{i}+\eta_{t}+\varepsilon_{it}\\ {\;\tau }_{it}=\gamma W\times \tau_{it}+\varepsilon_{it} \end{array}$$
(7)$$\begin{array}{ll} {\mathrm{lnY}}_{it}=\alpha_{it} & +\rho W{\mathrm{lnY}}_{it}+\beta_{1}GC_{it}+\beta_{2}lncapital_{it}+\beta_{3}lnlabor_{it}+\beta_{4}IS_{it}+\beta_{5}FDI_{it}\\ & +\beta_{6}STF_{it}+\;\beta_{7}\times GC_{it}+\beta_{8}Wlncapital_{it}+\beta_{9}lnlabor_{it}+\beta_{10}WIS_{it}\\ & +\beta_{11}WFDI_{it}+\beta_{12}WSTF_{it}+\beta_{13}ER_{it}+\beta_{14}GC_{it}\times ER_{it}+\beta_{15}WER_{it}\\ & +\beta_{16}WGC_{it}\times ER_{it}\varphi_{i}+\eta_{t}+\varepsilon_{it} \end{array}$$

If the regression coefficient of $GC_{it}\times ER_{it}$ is not significant, it indicates that the interaction does not significantly affect the development of GTI. If the regression coefficient of GC$GC_{it}$ is not significant or significantly negative and the regression coefficient of $GC_{it}\times ER_{it}$ is significant, it indicates that the effect of private investment on the development of GTI is not obvious and it needs to be combined with ER to produce a significant catalytic effect. If the regression coefficient of ER${\mathrm{ER}}_{\mathrm{it}}$ is not significant and the regression coefficient is significant, it indicates that GC can directly promote the development of GTI and does not need to rely on ER.

## 3. Results and Analysis

### 3.1. Descriptive Statistics

The descriptive statistics of all sample are showed in Table 1. The variables’ name, the sample sizes, and the mean values are shown in the first three columns of each group. The standard deviation, the minimum values, and the maximum values of each variable are shown in the last three columns. The average value of natural logarithm GTI (lnGTI) is 6.97 (min = 2.63, max = 10.07). The average of GC is 45.34 for the 300 observations. The standard deviation of GC is relatively high, which represents a large cross-sectional variation of areas ranging from 5.416 to 80.05. In other words, the overall level of GC is still not high, and large differences exist among provinces. In addition, the average ER is 1.389, and the standard deviation is 0.685 (min = 0.299, max = 4.111). Moreover, the distribution level of the remaining control variables is also basically reasonable, which is basically similar to the statistical results of existing research.

### 3.2. Correlation Analysis

The correlation matrix for ovariables is presented in Table 2. We find that the level of GC and ER are positively and significantly have relationship with GTI (0.535, 0.132 respectively). Hence, we can surmise that GC and ER could promote the development of green technology innovation. In addition, in order to test whether variables have multiple collinearity problems, the variance inflation factor (VIF) value is measured. When the maximum variance expansion factor, VIF = max, VIF_1_,…, VIF_n_, VIFition, in order to test whether variables have multiple collin. The calculations in this paper show that the maximum variance expansion factor is 3.95, and the mean variance expansion factor is 2.19, which is much less than 10, indicating that there is basically no collinearity between the variables in the model constructed in this paper.

### 3.3. Spatial and Temporal Evolution of GTI

The spatial difference characteristics of GTI level can be clearly seen from Figure 1. GTI in the eastern region in 2007–2016 has been in a dominant position, followed by that in the central region, the northeastern region and the western region.

As can be seen from the trend of change, the GTI level in four regions generally showed an upward trend from 2007 to 2016, with a marked increase in the development momentum of the eastern region. The reason is that the eastern region is more developed, superior in human resource, material, financial and technical conditions, industrial base, geographical location and green innovation ability. For the central and western regions, due to the gradual attention of the state to the imbalance of regional development and related policy preference, the universality and severity of pollution transfer to there in the course of development have attracted the attention of the central government, and corresponding policies and related measures have been adopted to govern and control the situation. Although these policies and measures have promoted the development of the central and western regions to a certain extent, the situation in the west remains serious. Due to the differences of the level of economic development, economic and industrial structure, and the ability of talent-technology transformation between regions, the level of GTI in the eastern, central, northeastern and western regions of China shows a decreasing trend.

### 3.4. Evolution of Spatial Autocorrelation of GTI

In this study, the spatial correlation test of GTI of different provinces and cities in 2007–2016 was carried out by using Geoda software (Dr. Luc Anselin and his team, Chicago, American). The spatial weight matrix is constructed by Queen Contiguity Weight, and the test results are given in Table 3.

Table 3 shows that the global Moran’s I index values of GTI from 2007 to 2016 are positive in all years and passed the statistical significance level at the level of 1%. This demonstrates that GTIs share a significant positive spatial correlation, characterized by spatial agglomeration and exert a significant spatial spillover effect on the surrounding area. Therefore, the influence effect of GC and ER on GTI should be further discussed by considering the spatial factors. From the Moran’s I index scatter diagrams (Figure 2 and Figure 3), it can also be observed that from 2007 to 2016, most of the GTIs in various regions focused on the third quadrant, showing a strong spatial autocorrelation relationship in space and a low-low agglomeration phenomenon. It indicates that the spatial distribution of GTI is not completely random, but shows an obvious heterogeneity. In short, it corresponds to objective facts to consider the spatial effect in the study on the effect model of GC and GTI.

### 3.5. Determination of Spatial Panel Model

Panel spatial econometric models mainly include the SLM, the SEM and the SDM, which are generally selected based on the Lagrange Multiplier (LM-lag and LM-err) of model residual and its robustness (Robust-LM-lag and Robust-LM-err). If the traditional panel judgment method is used to select the model, there may be some deviation. Thus, Elhorst put forward an improving method to improve the traditional LM method. In this paper, on the basis of Elhorst’s research methods [35], the models were tested with LM, and the results are shown in Table 4. As can be seen from Table 4, both the SLM and SEM models passed the test at a significant level of 1% under models of spatial fixed effects and spatial and time period fixed effects. In order to take the lag of the two models into account and avoid the influence of residual autocorrelation on the regression results, the SDM was further adopted for empirical analysis in this paper.

The spatial panel model is divided into fixed effect spatial panel model and stochastic effect spatial panel model. Since spatial econometricians tend to study the spatiotemporal data of geographically uninterrupted regions, fixed effect model is often used in the actual modeling process. The LM test results are listed in Table 4.

### 3.6. Model Estimation Results and Analysis

In order to select the model accurately, the SDM, the SLM and the SRM were estimated and tested, respectively. The estimation and test results are listed in Table 5.

With regard to GC, the regression coefficients of columns (1), (3) and (5) are 1.7726, 0.8891, 0.5690, respectively, all of which passed the test at a significant level of 1%, indicating that a single GC can promote GTI in the province. The regression coefficient of WGC in the SDM (column 5) is 0.7222. However, green credit did not pass the significance test, and did not play a certain role in the GTI of neighboring provinces. In view of the current situation in China, bank loans are still the main source and channel of capital for large and medium-sized enterprises. The development of GC by banks can effectively control or block the financing of enterprises with high energy consumption and high pollution, so as to promote the green transformation reform of these enterprises or their exit from the market. Meanwhile, GC can increase credit support for environmental protection and energy-saving industries, support the further development of green enterprises, and promote the optimization of industrial structure and the development of green innovation. In order to obtain more financial support, enterprises with high energy consumption and high pollution will try to transform to green industries and invest more in scientific innovation to enhance the level of green technology. However, the development of local GC will increase the difficulty of financing for local polluting enterprises and stimulate some enterprises to move to the surrounding areas, which may not promote GTI in the surrounding areas.

The regression coefficients of models after adding ER, namely, columns 2, 4 and 6, are 1.3123, 1.243 and 0.8048, respectively, all passed the test at a significant level of 1%, which shows that the enhancement of ER has a positive effect on the GTI in the province. The regression coefficient of W*ER is positive but not significant, indicating that in order to meet the requirements of government regulation, enterprises have to invest resources into GTI for maximizing pollution control in the limited range. With the increase of government regulation strength, the level of GTI of enterprises keeps rising; production costs keep declining; production efficiency keeps going up; and the green output of enterprises keeps increasing. However, ER in neighboring provinces has no significant impact on GTI of the province, which also indicates that ER needs to be strengthened, and its promotion effect on GTI has not been fully demonstrated.

For the interaction between GC and ER, the coefficient of GC * ER is −1.4936, which passes the test at the level of 1% significance, indicating that environmental regulation has a certain adjustment effect on the relationship between green credit and green technology innovation. The coefficient of WGC*ER is negative but not significant. The combination of environmental regulation and green credit has no impact on the green technology innovation of neighboring provinces.

Table 5 also suggests that *W* * dep.var. estimates in columns (1), (2), (5) and (6) are 0.0749, 0.0619, 0.589 and 0.5769, respectively, and are significant at the level of 1%, indicating a significant spatial positive correlation between GTIs of different provinces in China. With regard to control variables, as for STF and IS, which embody scientific and fiscal expenditure and industrial structure upgrading respectively, the direct effect coefficient is positive while the indirect effect (the influence on the neighboring provinces) coefficient is negative, and they all passed the test at the significant level of 1%. That is to say, the investment to science and technology and the development of the tertiary industry promoted the development of green technology of the province and had a negative impact on neighboring provinces. Similarly, for labor investment (lnlabor) and capital investment (lncapital), it has a positive impact on the province’s green technology innovation, but inhibits the development of green technology innovation in neighboring provinces. The impact of FDI on the development of green technology innovation in the province and neighboring provinces is not significant.

## 4. Conclusions

By depicting the dynamic evolution characteristics of GTI level in China, this paper focuses on the influence mechanism of GC and ER on GTI in China. Empirical analysis found that:

(1) According to the evolution over time, the level of GTI in China presents obvious spatial differences. The GTI in the eastern region in 2007–2016 has been in a dominant position, followed by that in the central region, the northeastern region and the western region. The GTI level in the four major regions showed a rising trend from 2007 to 2016, with that in the eastern region growing significantly faster than that in other regions. In addition, there is a positive spatial correlation between GTIs of different provinces in China, which indicates that GTIs of neighboring provinces interact with each other. Moreover, economic factors contribute to the spatial correlation. If the spatial characteristics in the variables are ignored and the classical measurement model of non-correlation and homogeneity hypothesis is used to depict the influence mechanism of GC on GTI, deviation of the empirical effect may occur.

(2) The above spatial econometrics analysis reveals that the level of GTI is not influenced and determined by a single element, but rather is the result of a multi-factor interaction. Besides, with the change of time, the results of interaction of various factors also differ. GC is conducive to promoting technology innovation of enterprises. The SLM, the SEM and the SDM all suggest that single GC has a significant role in promoting technology innovation of enterprises. The main reason is that GC can increase the research and development investment of enterprises and promote their technology innovation and product structure upgrading, thus positively influences technology innovation. However, the development of local GC will increase the difficulty of financing for local polluting enterprises, thus stimulating some enterprises to move to the surrounding areas, which may not promote GTI of the surrounding areas.

(3) Environmental regulation has a positive impact on the province’s green technology innovation. The results are similar to the paper of Lin and Yong [17]. They have studied the relationship between environmental regulation and technology innovation in three different regions of China based on data from 1985 to 2008 by using Co-integration and Granger Test. The results of their paper show that environmental regulation has a positive effect on technology innovation in these regions in the long term, but where their Granger causality relationship are different. So, under resource and environmental constraints, environmental policies which are oriented towards technology innovation should be made and enforced based on the facts of regional unbalanced economic growth in China. Although they considered regional imbalances, they did not consider spatial correlation. The spatial econometrics analysis reveals that ER can increase the research and development investment of enterprises and promote their technology innovation and product structure upgrading, thus positively influences technology innovation. However, the development of local ER will increase the difficulty of financing for local polluting enterprises, thus stimulating some enterprises to move to the surrounding areas, which may not promote GTI of the surrounding areas.

The combination of GC and ER play a significant role in the GTI of the province and its surrounding areas. GC should be integrated with other elements to promote GTI. The impact of GC on GTI of the province depends on the joint role of other factors, among which economic development, investment in science and technology, upgrading of industrial structure and, etc. play significant roles in promoting GTI of the province. Nevertheless, the development of the factors of neighboring provinces cannot promote the development of GTI of the province.

Through the conclusions drawn in this paper, the following policy implications can be obtained: 

(1) Sound investment environment should be created to promote effective supply of private investment. The empirical conclusion shows that green credit can stimulate technology innovation and the innovation vitality of enterprises. Therefore, in order to promote the stable and healthy development of GC and give full play to government policies, financial restraint mechanism and incentive mechanism should be established by relevant departments. On the basis of the restraint mechanism, the supervision and assessment of environmental performance should be strengthened. Commercial banks that fail to meet the requirements should be held to account and be heavily punished so as to prevent them loaning to the industries with high energy consumption and heavy pollution. Based on the incentive mechanism, corresponding financial subsidies should be provided to commercial banks that conform to GC policies to a certain degree, and at the same time loan with discounted interest should be provided to green development projects. In addition, an environmental information sharing platform can be established to realize tripartite information sharing between regulatory departments, commercial banks and enterprises. Meanwhile, GC development reports should be issued on a regular basis to improve the market competitiveness of outstanding banks and enterprises. The conclusions are consistent with the study of Owen et al. (2018) [36]. They explore the role of public sector support for grant, equity, debt and new forms of crowd funding finance in higher and lower income countries. They conclude that a finance ecosystem approach is required that ensures complementary forms of finance for low carbon investment are connected at local, national and international scales, alongside support to build entrepreneurial skills and investment readiness. There is also a need for better evidence of the role of public sector support and where there is greatest impact on climate change.

(2) An environmental forcing mechanism can be established to guide the healthy development of GC. From the conclusion of the empirical analysis, it can be seen that environmental regulation has a positive impact on local GTI, while the combination of the two fails to play an effective role. Therefore, practical and feasible policies need to be formulated to encourage investment of GC to high-end green industries. Meanwhile, an environmental forcing mechanism should be established to guide the healthy development of green investment and promote green technology innovation of enterprises. (3) Regional synergy should be promoted and given full play to promote GTI of enterprises. The empirical conclusion suggests that GTI can be influenced by many factors and has an interactive influence on different regions. Therefore, at the national level, a regional coordination mechanism of environmental regulation should be established in an integrated manner. Furthermore, the positive spillover effect of GTI between the province and neighboring provinces should be given full play to form a pattern of common development between regions. In order to promote the development of GTI, more funds should be invested in research and development [37], and the advantage of ER and research and development investment integration should be fully used [38]. In addition, government financial expenditure should be increased to guarantee fund supply, improve the degree of industrial concentration, and promote the transformation and upgrading of enterprises [39].

## Figures and Tables

**Figure 1 ijerph-16-03027-f001:**
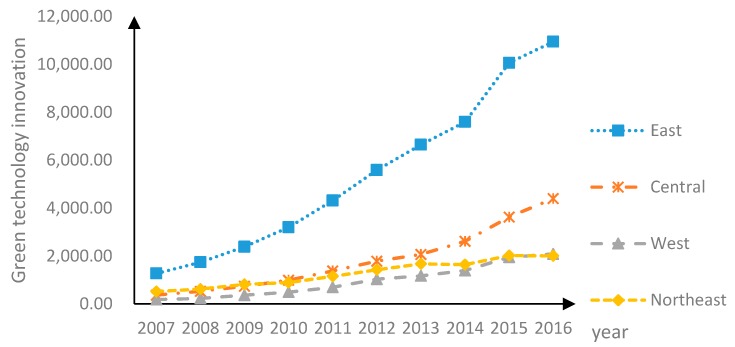
Change trend of GTI level in China in 2007–2016.

**Figure 2 ijerph-16-03027-f002:**
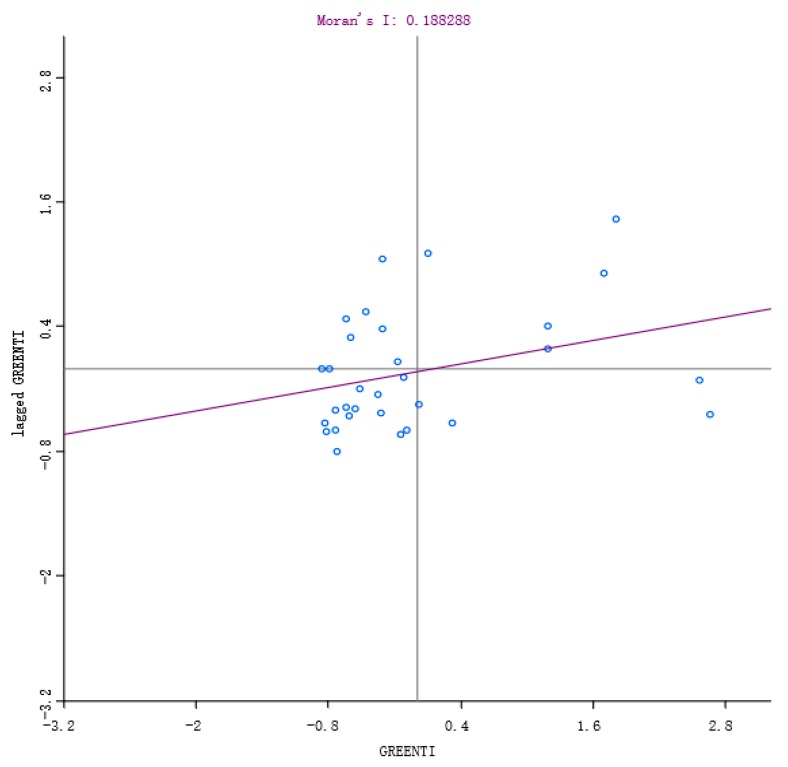
Scatter diagram of Moran’s I in 2007.

**Figure 3 ijerph-16-03027-f003:**
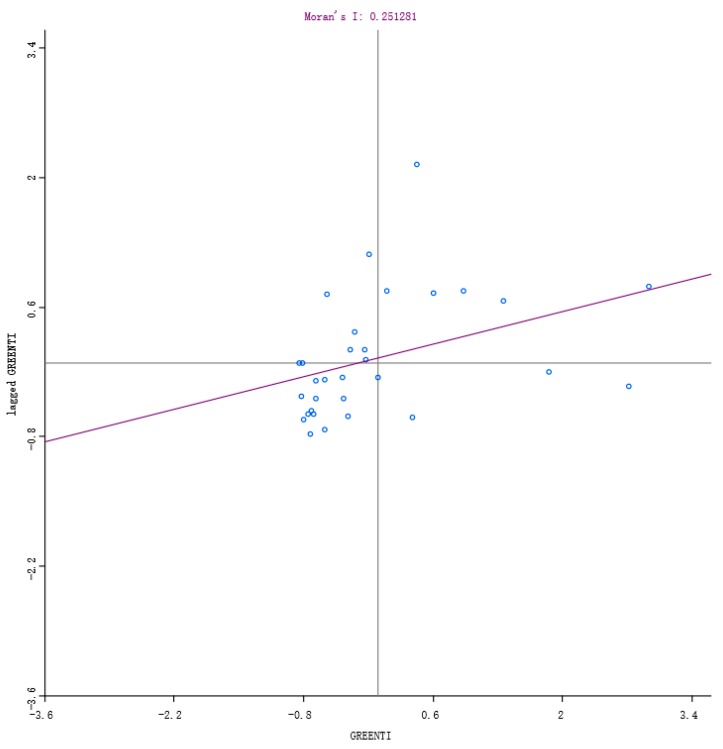
Scatter diagram of Moran’s I in 2016.

**Table 1 ijerph-16-03027-t001:** Descriptive statistics.

Variable	Observ Ations	Mean	Standard Deviation	Min	Max
lnGTI (average value of natural logarithm)	300	6.97	1.485	2.63	10.07
GC (green credit)	300	45.34	14.54	5.416	80.05
ER (environmental regulation)	300	1.389	0.685	0.299	4.111
IS (industrial structure)	300	42.30	9.085	28.60	80.23
FDI (foreign direct investment)	300	2.323	1.791	0.0387	8.190
STF (technological fiscal expenditure)	300	3.132	6.540	0	39.90
lnlabor	300	10.40	1.370	5.682	13.03
lncapital	300	0.743	0.382	0.240	3.100

**Table 2 ijerph-16-03027-t002:** Correlation Analysis.

	GTI	GC	ER	IS	FDI	STF	lnlabor	lncapital
GTI	1							
GC	0.535 ***	1						
ER	0.132 **	0.178 ***	1					
IS	0.493 ***	0.207 ***	0.00400	1				
FDI	−0.170 ***	−0.348 ***	−0.210 ***	−0.226 ***	1			
STF	0.156 ***	0.096 *	0.187 ***	0.108 *	0.0800	1		
lnlabor	0.630 ***	0.682 ***	0.246 ***	0.153 ***	0.284 ***	0.169 ***	1	
lncapital	0.226 ***	0.237 ***	0.429 ***	0.245 ***	0.358 ***	0.107 *	0.331 ***	1

Note: *, **, and *** indicate statistical significance at the 10%, 5%, and 1% levels, respectively.

**Table 3 ijerph-16-03027-t003:** Moran’s I index of GTI of 30 provinces and cities in China in 2007–2016.

Year	Moran’I	*z*-Value	*p*-Value	Year	Moran’I	*z*-Value	*p*-Value
2007	0.1883	1.9463	0.038	2012	0.2616	2.7793	0.011
2008	0.1949	2.02.4	0.04	2013	0.2054	2.269	0.025
2009	0.2205	2.3215	0.022	2014	0.2195	2.4333	0.024
2010	0.2429	2.4975	0.019	2015	0.2682	2.8859	0.006
2011	0.2578	2.6017	0.018	2016	0.2513	2.5347	0.017

**Table 4 ijerph-16-03027-t004:** LM test of spatial panel model.

LM Test	No Fixed Effects		Time Period Fixed Effects		Spatial Fixed Effects		Spatial and Time Period Fixed Effects	
	*z*	*p*	*z*	*p*	*z*	*p*	*z*	*p*
LM test no spatial lag	775.8084	0	0.4996	0.480	4.9141	0.027	9.8991	0.02
Robust LM test no spatial lag	1440.0568	0	15.3422	0	106.6592	0	14.7746	0
LM test no spatial error	104.2632	0	83.8954	0	279.2499	0	83.9936	0

**Table 5 ijerph-16-03027-t005:** Spatial panel model estimation results of influence factors of GTI in China.

	SLM		SEM		SDM	
	(1)	(2)	(3)	(4)	(5)	(6)
GC	1.7726 ***	1.9043 ***	0.8891 **	2.1819 ***	0.5690 **	1.3590 **
	(5.2543)	(−3.2877)	(2.4358)	(−3.4950)	(2.6082)	(−2.1808)
IS	0.0316 ***	0.0397 ***	0.0294 ***	0.0414 ***	0.0277	0.0360 ***
	(7.9720)	(10.6520)	(6.7363)	(9.5691)	(6.4763) ***	(8.0066)
FDI	−0.0268	−0.0270	−0.0298	−0.0411	−0.0218	−0.0316
	(−1.1416)	(−1.2746)	(−1.0687)	(−1.5834)	(−0.7944)	(−1.1961)
lnlabor	0.5572 **	0.6570 ***	0.7643 ***	0.7619 ***	0.8367 ***	0.8374 ***
	(18.0589)	(21.4885)	(23.1448)	(24.7912)	(25.1648)	(26.2567)
lncapital	0.0101 ***	0.432189 ***	0.5344 ***	0.6842 ***	0.7215 ***	0.8497 ***
	(0.1052)	(4.1837)	(4.3207)	(5.6613)	(5.6573)	(6.7380)
STF	0.02547 ***	0.0188 ***	−0.0188 **	−0.0017	0.0291 ***	0.0203 ***
	(4.4506)	(3.6009)	(−2.4299)	(-0.2399)	(−3.8597)	(−2.5845)
ER		1.3123 ***		1.2434 ***		0.8048 ***
		(−8.2922)		(−6.8278)		(−4.1898)
GC*ER		−2.4493 ***		−2.3884 ***		−1.4936 ***
		(7.2825)		(5.9317)		(3.5250)
WGC					0.7222	1.1568
					(1.3737)	(1.2077)
WIS					−0.0221 ***	−0.0261 ***
					(−3.5522)	(−4.1202)
WFDI					0.0126	0.0224
					(0.3480)	(0.6378)
Wlnlabor					−0.649836 ***	−0.5946 **
					(−13.2990)	(−11.0143)
Wlncapital					−0.8666 ***	−0.7800 ***
					(−5.1084)	(−4.2823)
WSTF					−0.0395	−0.0275 ***
					4.4770 ***	(3.0603)
WER						0.2876
						(1.0135)
WGC*ER						−0.5998
						(−1.0279)
W * dep.var.	0.0749 **	0.0619			0.5899 ***	0.5769 ***
	(1.7767)	(1.6197)			(13.1189)	(12.4160)
Spatial autoregressive parameter (spat.aut.)			0.76698 ***	0.6719 ***		
			(27.1054)	(17.6499)		
R-squared	0.8961	0.9154	0.8369	0.8979	0.9485	0.9527
sigma^2^	0.3786	0.3083	0.2210	0.2085	0.1876	0.1724
log-likelihood	−289.5560	−257.6234	−231.8897	−210.5892	−201.3879	−187.1895

Note: t statistics are enclosed in parentheses. *, **, and *** indicate statistical significance at the 10%, 5%, and 1% levels, respectively.

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
