# Peer review of "Spatial Effects of Environmental Regulation and Green Credits on Green Technology Innovation under Low-Carbon Economy Background Conditions"

_ijerph, 2019, doi:10.3390/ijerph16173027_

Round 1
Reviewer 1 Report
Dear Authors, please use others countries experience to compare yours research results in more comprehensive discussions.
Author Response
Response to Reviewer
Quan Guo, Min Zhou, Nana Liu and Yaoyu Wang
At first, we appreciate editors’ and reviewers’ valuable comments and suggestions which help us improve the paper significantly.
Response to Reviewer 1 Comments
Point 1: Dear Authors, please use others countries experience to compare yours research results in more comprehensive discussions.
Response1: Thanks for the suggestion, and we think your advice is very important. We have added some similar study of China and others countries experience to compare our research results and conclusions in the part of conclusions and discussion.
“(3) Environmental regulation has a positive impact on the province's green technology innovation. The results are similar to the paper of Lin and Yong(2010)[35]. They have studied the relationship between environmental regulation and technology innovation in three different regions of China based on data from 1985 to 2008 by using Co-integration and Granger Test. The results of their paper show that envirnomental regulation has a positive effect on technology innovation in these regions in the long term, but where their Granger causality relationship are different. So, under resource and environmental constraints, environmental policies which are oriented towards technology innovation should be made and enforced based on the facts of regional unbalanced economic growth in China. Although they considered regional imbalances, they did not consider spatial correlation. The spatial econometrics analysis reveals that ER can increase the research and development investment of enterprises and promote their technology innovation and product structure upgrading, thus positively influences technology innovation. However, the development of local ER will increase the difficulty of financing for local polluting enterprises, thus stimulating some enterprises to move to the surrounding areas, which may not promote GTI of the surrounding areas.
The combination of GC and ER play a significant role in the GTI of the province and its surrounding areas. …”
‘’ (1) Sound investment environment should be created to promote effective supply of private investment. …The conclusions are consistent with the study of Owen et al.(2018)[36]. They explore the role of public sector support for grant, equity, debt and new forms of crowd funding finance in higher and lower income countries. They conclude that a finance ecosystem approach is required that ensures complementary forms of finance for low carbon investment are connected at local, national and international scales, alongside support to build entrepreneurial skills and investment readiness. There is also a need for better evidence of the role of public sector support and where there is greatest impact on climate change. ’’
On behalf of co-authors, we thank you very much for giving us an opportunity to revise our manuscript, we appreciate you very much for your positive and constructive comments and suggestions on our manuscript. We are also very grateful to you for your reference. In the future research, we will be more rigorous and careful.

Reviewer 2 Report
This study, based on the data of green credit (GC), environmental regulation (ER) and green technology innovation (GTI) in 30 provinces and cities of China from 2007 to 2016, investigated the relationship between GC, ER and GTI. The paper is well structured and suitably documented, showing a proper empirical approach. Nevertheless, several shortcomings should be fixed, as detailed below.
1. I think the conclusion of policy implications should not include in the abstract.
2. The Introduction should clearly state the contribution of the study to the literature. What are exactly the novelties? The “three wastes” and ““13th Five-Year Plan” appears suddenly on line 43 and 53. What is exactly the meaning? The reader is left to guess.
3. The literature review is too compact. Some literatures are available regarding the role of green credit, environmental regulation and green technology innovation, such as, Minghua, & Yongzhong (2011), Feng and Weng (2018), Guo, Xia, Zhang, & Zhang (2018), Liu, Wang, & Cai (2018), Hu, Wang, Lian, & Huang (2018), Sun, Wang, Yin, & Zhang (2019). The authors should be added to the manuscript.
4. Control variables used in the model are insufficient in order to control for green technology innovation. It is a problem of data availability, but it is an important problem, since the paper is explaining green technology innovation and there are many factors affecting on this variable that are not present in the empirical model. Some control variables could raise problems. On the one hand, whether the measure of lnGDP, UR, IS, FDI and STF have a relationship among each other, and it is especially relevant in panel data models.
5. The methodological section presents well the steps regarding the relationship between GC, ER and GTI. However, this study seems as an exercise by putting the data of GC, ER and GTI in the software to get the results. There is nothing new in the methodology.
6. The Chinese word “和” ( line 256) appears in the study, it seems that this study is not rigorous enough.
7. The third section reveals the empirical outcomes. The results show that the combination of GC and ER exerts no significant effect on GTI of the province and neighboring provinces. Table 3 shows that most of the variables are not significant; whether the result is reliable and accurate. I suggest the author(s) to provide descriptive statistics of the selected variables, as well as the correlations.
References
Feng, W. E. I., FENG, B. M., Jian, W. A. N. G., & Qin, Y. Y. (2017). Research on Green Innovation of Traditional Manufacturing Enterprises in the Context of Supply Side Structural Reform. DEStech Transactions on Economics, Business and Management, (icem) 388-291.
Feng, Z., & Chen, W. (2018). Environmental regulation, green innovation, and industrial green development: An empirical analysis based on the Spatial Durbin model. Sustainability, 10(1), 223.
Guo, Y., Xia, X., Zhang, S., & Zhang, D. (2018). Environmental regulation, government R&D funding and green technology innovation: evidence from China provincial data. Sustainability, 10(4), 940.
Hu, J., Wang, Z., Lian, Y., & Huang, Q. (2018). Environmental regulation, foreign direct investment and green technological progress—Evidence from Chinese manufacturing industries. International journal of environmental research and public health, 15(2), 221.
Liu, X., Wang, E., & Cai, D. (2018). Environmental Regulation and Corporate Financing—Quasi-Natural Experiment Evidence from China. Sustainability, 10(11), 4028.
Minghua, L., & Yongzhong, Y. (2011). Environmental regulation and technology innovation: Evidence from China. Energy Procedia, 5, 572-576.
Sun, J., Wang, F., Yin, H., & Zhang, B. (2019). Money Talks: The Environmental Impact of China's Green Credit Policy. Journal of Policy Analysis and Management, 38(3), 653–680.
Reviewer 3 Report
I would be happy to recommend the paper for your journal with some minor revisions.
The abstract should be shorten. It spends too much space to say low carbon production is important. It is too long. Introduction part should be improved. It talks a lot about the how important the environmental issue is. This is well known. Rather, you are talking about the problems in China, therefore, you should give a little bit more information about China. The paper claims that the data are collected from 30 provinces. Readers are not clear Beijing, Tianjin, Shanghai and Chongqing are counted as provinces. You should also give an explanation about why Xizang (Tibet) is excluded. Some English bugs are found. (The overall English level of this paper is very good, clear). You should fix them.Author Response
Response to Reviewer
Quan Guo, Min Zhou, Nana Liu and Yaoyu Wang
At first, we appreciate editors’ and reviewers’ valuable comments and suggestions which help us improve the paper significantly.
Response to Reviewer 3 Comments
Point 1:I would be happy to recommend the paper for your journal with some minor revisions.
The abstract should be shorten. It spends too much space to say low carbon production is important. It is too long.
Response1: Thanks for the suggestion. According to this point, we have rewrite the abstract, and the revised abstract are as follows: ” Abstract: Based on the data of green credit (GC), environmental regulation (ER) and green technology innovation (GTI) in 30 provinces and cities of China from 2007 to 2016, this study investigated the relationship between green credit and green technology innovation development and analyzed the adjustment effect of ER on GC to promote GTI using Geoda and Matlab2016 software, so as to further guide and encourage GC. The results show that GTI in 30 provinces and municipalities in China has a significant spatial agglomeration effect. Single GC plays a certain role in promoting local technology innovation, but it fails to influences the surrounding areas. Environmental regulation has a certain regulatory effect on the relationship between green credit and green technology innovation in the province but also fails to influences the surrounding areas.”
Point 2: Introduction part should be improved. It talks a lot about the how important the environmental issue is. This is well known. Rather, you are talking about the problems in China, therefore, you should give a little bit more information about China.
Response2: Thank you for your advice, and we think your advice is very important. We haved added more information on environment problems in China on lines 41-74.
“Given the social and economic development, as well as policy advocacy, China has experienced a rapid and sudden increase in economic development in the past three decades. Although economy in China is rapidly developing, numerous problems have also occurred. The extensive growth of China’s economy, affects the ecology and environment and causes natural resource depletion and population explosion; these phenomena potentially hinder “substantial development” [2]. According to the joint report on the Global Environmental Performance Index (GEPI) issued by Yale University Center for Environmental Law and Policy (YCELP), Columbia University Center for International Geoscience Information Network (CIESIN)[3] and the World Economic Forum (WEF), China ranked 94 (40th percentile), 105 (45th percentile),121 (43th percentile), 116 (17th percentile), 118 (61th percentile) and 109 (72th percentile) in 2006, 2008,2010, 2012, 2014 and 2016. The extensive industrial development model has made China fall into the “environmental pollution–economic development” circle. In this regard, green technology innovation is the key to get out of this “strange circle”[4].
Point 3:The paper claims that the data are collected from 30 provinces. Readers are not clear Beijing, Tianjin, Shanghai and Chongqing are counted as provinces. You should also give an explanation about why Xizang (Tibet) is excluded.
Response3: Thanks for the suggestion. We haved provided a more detailed explanation of the choices for the provinces.
“China has 34 provincial-level administration regions, of which 30 provinces and cities (except Tibet, Hong Kong, Macao, and Taiwan) from 2007 to 2016 are investigated in this study, considering the continuity of index and the availability of data. These 30 provincial-level administration regions can be divided into four groups, namely, eastern, northeastern, central, and western regions. The eastern region comprises 10 administrative areas, including three out of the four municipalities in China, namely, Beijing, Shanghai, and Tianjin. The other seven administrative areas include Fujian, Guangdong, Hainan, Hebei, Jiangsu, Shandong, and Zhejiang Provinces. The northeastern region includes Heilongjiang, Jilin, and Liaoning Provinces, whereas the central region includes Anhui, Henan, Hubei, Hunan, Jiangxi, and Shanxi Provinces. The western region includes four autonomous regions, one municipality (i.e., Chongqing), and six provinces. The autonomous regions are Guangxi, Inner Mongolia, Ningxia, and Xinjiang, and the provinces are Gansu, Guizhou, Qinghai, Shaanxi, Sichuan, and Yunnan[25].”
Point 4: Some English bugs are found. (The overall English level of this paper is very good, clear). You should fix them.
Response4: We are very grateful to you for your comments, and very sorry for our careless. We have corrected the English mistakes.
On behalf of co-authors, we thank you very much for giving us an opportunity to revise our manuscript, we appreciate you very much for your positive and constructive comments and suggestions on our manuscript. We are also very grateful to you for your reference. In the future research, we will be more rigorous and careful.
